# Do childhood socioeconomic circumstances moderate the association between childhood cognitive ability and all-cause mortality across the life course? Prospective observational study of the 36-day sample of the Scottish Mental Survey 1947

Matthew Henry Iveson [1,2], Drew Altschul,[1,2] Ian Deary[2,3]

For numbered affiliations see end of article.

**Correspondence to**
Dr Matthew Henry Iveson;
Matthew.Iveson@ed.ac.uk

## ABSTRACT

**Background** There is growing evidence that higher childhood cognitive ability predicts lower all-cause mortality risk across the life course. Whereas this association does not appear to be mediated by childhood socioeconomic circumstances, it is unclear whether socioeconomic circumstances moderate this association.

**Methods** The moderating role of childhood socioeconomic circumstances was assessed in 5318 members of the 36-day sample of the Scottish Mental Survey 1947. Univariate, sex-adjusted and age-adjusted, and mutually adjusted Cox models predicting all-cause mortality risk up to age 79 years were created using childhood IQ scores and childhood social class as predictors. Moderation was assessed by adding an interaction term between IQ scores and social class and comparing model fit.

**Results** An SD advantage in childhood IQ scores (HR=0.83, 95% CI 0.79 to 0.86, p<0.001) and a single-class advantage in childhood social class (HR=0.92, 95% CI 0.88 to 0.97, p<0.001) independently predicted lower mortality risk. Adding the IQ–social class interaction effect did not improve model fit ($\chi^2\Delta$=1.36, p=0.24), and the interaction effect did not predict mortality risk (HR=1.03, 95% CI 0.98 to 1.07, p=0.25).

**Conclusions** The present study demonstrated that the association between higher childhood cognitive ability and lower all-cause mortality risk is not conditional on childhood social class. Whereas other measures of socioeconomic circumstances may play a moderating role, these findings suggest that the benefits of higher childhood cognitive ability for longevity apply regardless of the material socioeconomic circumstances experienced in childhood.

## INTRODUCTION

Several studies have demonstrated an association between higher cognitive ability and lower risk of all-cause mortality.[1–3] This association stretches across the life course,

### Strengths and limitations of this study

► All-cause mortality risk was examined in a large birth cohort over an entire life course.
► The study used contemporaneously measured indicators of both childhood socioeconomic circumstances and childhood cognitive ability.
► The present study used a more fine-grained measure of childhood socioeconomic circumstances than previous research that better captures material resources in early life.
► Use of father's social class as a measure of childhood socioeconomic circumstances means potential miscoding of those whose father had a missing occupation.
► The particular social, economic and educational changes experienced by these older adults may limit the generalisability to younger cohorts.

with childhood cognitive ability predicting mortality risk up to much later in life.[4–8] There have been several attempts to model these associations while accounting for a suite of demographic and psychosocial factors as covariates. Perhaps being the most common covariate to be examined, some studies have reported that childhood socioeconomic circumstances mediate or confound the association between early-life cognitive ability and all-cause mortality risk either very slightly or not at all.[6 7 9–11] However, the commonality between these studies has been the tendency to focus on examining only confounding and mediation effects while ignoring potential moderation of the cognitive ability–mortality association by socioeconomic circumstances. Whereas a mediator explains the association

between two other variables (ie, between childhood cognitive ability and all-cause mortality), a moderator influences the strength of the association between the variables. That is, the strength of the association between childhood cognitive ability and all-cause mortality risk may be conditional on socioeconomic circumstances; a strong association may be observed for some levels of socioeconomic circumstances, but a weaker (or no) association may be observed for others.

Only a few studies have considered the moderating role of socioeconomic circumstances in the association between cognitive ability and all-cause mortality. In a study using the Helsinki Birth Cohort, higher cognitive ability at age 20 years was associated with lower risk of mortality in later life, and this association was stronger among those from middle-class than manual–occupational-class families.[12] Similar moderation has been shown in the US National Longitudinal Survey of Youth 1979, although using parental education instead of occupational class.[13] In that study, the association between higher cognitive ability in young adulthood and lower all-cause mortality risk was significant only for those whose parents had high school qualifications or above, and not for those whose parents had not completed high school. In these studies, better socioeconomic circumstances is thought to allow access to the lifestyle and resources that enable higher cognitive abilities to impact on health, such as through healthier diet and more physical activity.[12] At the same time, lower socioeconomic circumstances may over-ride much of the protective contribution of higher cognitive ability through exposure to other associated risk factors.[13] However, these two studies are limited in terms of their samples—such as examining only men[12]—and in terms of their mortality follow-up—such as covering only mid-life mortality.[13] Given the established sex differences in mortality risk,[1 5 13] it is important to assess the moderation of the childhood cognitive ability–mortality risk association in a more population-representative sample. Furthermore, given that mortality risk increases with age and that childhood cognitive ability and socioeconomic circumstances both predict mortality risk in older adults,[6 8] a longer follow-up is necessary to fully capture the contribution of a moderation effect. These studies are also limited by their use of early adulthood cognitive ability as a predictor of mortality, whereas many studies report a strong association between cognitive ability in childhood and mortality across the life course.[4–7] Cognitive ability measured in childhood has the advantage of being upstream of many of the factors that may confound the contribution of later cognitive ability, such as education.[14] Although there appears to be little evidence that childhood socioeconomic circumstances mediate the association between childhood cognitive ability and all-cause mortality risk, their role as a moderator of this association remains unclear.

In the present study, we examine the role of childhood socioeconomic circumstances as a possible moderator of the association between childhood cognitive ability and

all-cause mortality risk across the life course. In particular, we use a representative sample of men and women born in Scotland in 1936 and whose cognitive ability was assessed as part of the Scottish Mental Survey 1947 and who were followed up for mortality up to 2015. Importantly, an association between childhood cognitive ability and all-cause mortality has already been established in this sample (HR=0.80, 95% CI 0.78 to 0.81[6 14 15 6 14 15]) and in a subsample of these individuals (HR=0.76, 95% CI 0.68 to 0.84[7]); however, potential moderation by childhood socioeconomic circumstances has not been investigated. Given previous observations using adult cognitive ability,[12 13] we predicted that childhood socioeconomic circumstances would significantly moderate the association between childhood cognitive ability and all-cause mortality, with stronger associations in those whose fathers had less manual-based occupations.

## METHODS
### Sample
The sample consisted of 6291 individuals from the 36-day sample of the Scottish Mental Survey 1947.[16] These individuals, born on 1 of 36 birth dates in 1936 (the first 3 days of each month), took part in a nationwide test of cognitive ability on 4 June 1947 (age 11 years). They have been shown to be representative of the wider Scottish Mental Survey 1947 cohort in terms of childhood cognitive ability and socioeconomic conditions.[15 16] The 36-day sample was subsequently followed up across the life course, both in person and remotely using routinely collected data.[14 16 17] In particular, these individuals were traced in national migration and death records.

To create the analytical sample (n=5318), we removed individuals with missing childhood cognitive ability (n=575), missing childhood socioeconomic circumstances (n=130), missing vitality status (n=312) or missing date of death (if deceased, n=12). Note that these individuals could have more than one key variable missing. Individuals who emigrated after completing the Scottish Mental Survey 1947 but before the end of the surveillance period were retained and censored appropriately.

### Measures
#### Cognitive ability
Childhood cognitive ability around the age of 11 years was assessed using the Moray House Test (MHT) No. 12 test of intelligence (MHT score 15) administered in groups as part of the Scottish Mental Survey 1947. The MHT consists of 71 items including verbal reasoning, word classification, mixed sentences, same–opposites, analogies, proverbs, arithmetic, cipher decoding, following directions, proverbs, spatial problem-solving and practical items. Further details of the MHT and its administration are provided elsewhere.[18] Raw MHT scores (ranging from 0 to 76) were converted to IQ-type score (mean=100, SD=15) and then standardised (z-transformed).

## Socioeconomic circumstances

The occupational social class of the father was used as a proxy of childhood socioeconomic circumstances at 11 years of age. Father's occupation was recorded in 1947 as part of the Sociological Schedule—a planned follow-up to the Scottish Mental Survey 1947, completed mostly by head teachers and school medical staff.[16] Note that this was based on their current or most recent occupation. The most recent occupation of fathers who were without occupations, were unemployed, or were absent or deceased was recorded where possible; otherwise occupation was recorded as missing. Father's occupation was later categorised into one of five occupational social classes according to the 1950 United Kingdom's Classification Index.[19] The social classes were reversed so that higher-numbered classes represented more professional occupations: (1) unskilled, (2) partly skilled, (3) skilled, (4) managerial and (5) professional.

## Mortality

All-cause mortality (status and date of death) was obtained from National Records of Scotland death records covering 1947–2015. The linkage and extraction of death records have been described in detail elsewhere.[7 17] Individuals were censored at the end of mortality surveillance (30 November 2015). Survival time was calculated as the number of days between 4 June 1947 and death or censoring date, as appropriate.

## Analyses

We first described the demographic characteristics of the analytical sample, split by mortality status, before examining cohort selection effects by comparing excluded and retained individuals.

We then conducted survival analyses using Cox proportional hazards regression. All models incorporated inverse probability weights to adjust for sample selection effects (sex, age in days at Scottish Mental Survey 1947, z-transformed IQ scores and father's occupational social class as predictors of analytical sample membership; 36-day sample: weight mean=1.98, SD=4.14; analytical sample: weight mean=1.05, SD=0.01). HRs were estimated to indicate the proportional change in all-cause mortality risk associated with a unit change in each predictor. As in previous work, individual models were constructed to assess the association between standardised IQ scores and reversed father's occupational social class (separately) and mortality risk. Consistent with previous work,[13] models were then adjusted for two potential confounders: sex to account for well-established sex differences in mortality risk and age in days at Scottish Mental Survey 1947 to account for small variations in age. A mutually adjusted model was then created, including cognitive ability, social class and sex. Novel to this study, we further constructed a model additionally including an interaction term between cognitive ability and social class. In addition to examining predictors within a given model, the effect of adding a given predictor was assessed by comparing the fit of successive models (eg, mutually adjusted vs mutually adjusted and moderated) using $\chi^2$ tests. Univariate survival models, including standardised IQ score, were then created for each social class in order to better assess interaction effects (or lack thereof).

Analysis was conducted in R[20] using RStudio,[21] and with the 'psych'[22] and 'survival'[23] packages.

The analysis plan was preregistered prior to the data being analysed (https://osf.io/u8y9n/). The only change to the planned analysis was the treatment of father's occupational social class as numerical (not centred or transformed) during the survival analyses, rather than ordinal. This was done because use of the ordinal variable violated the proportional hazards assumption, and use of a numeric variable provided a similar model fit to a categorical variable.

## Patient and public involvement

This study was based on analysis of previously collected and anonymised research data and routinely collected health data. As such, neither participants nor the public were involved in the design or planning of this study.

## RESULTS

Descriptive statistics for the analytical sample, split by father's occupational social class, are shown in table 1. Mortality status (p<0.001), survival times (p<0.001) and MHT scores (p<0.001) significantly differed between social classes, with a higher proportion of those in higher social classes being alive at censor date (professional=69.57%, unskilled=55.64%), surviving longer (professional vs unskilled mean difference=3.74 years) and demonstrating higher MHT scores (professional vs unskilled mean difference=16.97). Note, however, that the mean survival time of individuals with fathers from skilled occupations was lower than those with fathers from semiskilled occupations, despite a lower proportion of deceased individuals. Furthermore, there were relatively few individuals with a father from a professional occupation. MHT scores were significantly and positively correlated with both father's occupational social class (Spearman's rho=0.19, p<0.001) and survival time (Spearman's rho=0.07, p<0.001). Likewise, social class was significantly and positively correlated with survival time (Spearman's rho=0.04, p<0.01).

We then examined selection effects by comparing MHT scores and social class between the analytical sample and those removed due to having missing key variables. Note that this comparison necessarily included only those with at least one complete variable (eg, a complete MHT score but missing father's social class). Removed individuals had significantly lower MHT score (removed mean=34.42, t(454)=−2.22, p<0.05), but the samples did not significantly differ in terms of social class ($\chi^2$=9.30, p=0.05). This trend has been reported elsewhere[24 25] and may indicate that individuals with lower childhood cognitive ability were more likely to have long-term unemployed

**Table 1** Descriptive statistics for the analytical sample, split by father's occupational social class

| | Unskilled (n=1073) | Semiskilled (n=1026) | Skilled (n=2701) | Intermediate (n=472) | Professional (n=46) | Total (N=5318) |
|---|---|---|---|---|---|---|
| **Sex** | | | | | | |
| Male, n (%) | 546 (50.89) | 530 (51.66) | 1332 (49.32) | 239 (50.64) | 21 (45.65) | 2668 (50.17) |
| Female, n (%) | 527 (49.11) | 496 (48.34) | 1369 (50.68) | 233 (49.36) | 25 (54.35) | 2650 (49.83) |
| **Moray House Test of Intelligence Score (range=0–74)** | | | | | | |
| Mean (SD) | 31.88 (15.39) | 33.67 (15.57) | 38.21 (15.18) | 39.74 (15.91) | 48.85 (12.69) | 36.29 (15.65) |
| **Mortality status** | | | | | | |
| Alive, n (%) | 597 (55.64) | 632 (61.60) | 1703 (63.05) | 321 (68.01) | 32 (69.57) | 3285 (61.77) |
| Dead, n (%) | 476 (44.36) | 394 (38.40) | 998 (36.95) | 151 (31.99) | 14 (30.43) | 2033 (38.23) |
| **Time from SMS1947 to death/censor (years)** | | | | | | |
| Mean (SD) | 57.14 (16.82) | 57.78 (17.01) | 56.79 (18.16) | 59.96 (15.65) | 60.88 (16.62) | 57.37 (17.47) |

SMS1947, Scottish Mental Survey 1947.

fathers who could not be ascribed an occupational social class. Note, however, that differences in MHT score were small; removed individuals scored on average 1.87 points fewer on the MHT.

Returning to the analytical sample, we next compared descriptive statistics between those alive (n=3285) and those dead (n=2033) (table 1). In terms of sex, proportionately more men had died than women (p<0.001). Those who died exhibited significantly lower MHT scores (M=33.40, SD=15.89) and tended to be from more manual social classes (unskilled=23.41%) than those who survived (MHT: M=38.07, SD=15.23, p<0.001; social class: unskilled=18.17%, p<0.001).

We then conducted survival analyses aimed at examining whether social class moderated the association between childhood cognitive ability and all-cause mortality risk. Results from the Schoenfeld test and visual inspection of the scaled Schoenfeld residuals supported the proportional hazard assumption for all predictors in each model (once ordinal social class was replaced with numerical social class). Kaplan-Meier curves showing the association between IQ and survival within each social class are shown in figure 1.

Before examining moderation effects, we tested the independent associations of age-11 IQ and father's social class with all-cause mortality risk, as in previous work. In the univariate models, a 1 SD advantage in age-11 IQ predicted a 19% lower mortality risk over time (table 2). A one-class advantage in father's social class predicted an 11% lower mortality risk over time. The strength of these associations was similar after adjusting for sex and age at Scottish Mental Survey 1947, though adding sex and age significantly improved the fit of each model (p values<0.001). Likewise, mutual adjustment (including sex and age) did not wholly or partly attenuate any one association with mortality risk and significantly improved model fit versus both sex-adjusted models (ps<0.001). A

1 SD advantage in age-11 IQ and a one-class advantage in father's social class were both independently associated with small–moderate reductions in all-cause mortality risk (18% and 7%, respectively; table 2). Note that bias analyses suggested that any unmeasured confounder would require a strong association with the predictors and mortality risk in order to fully account for their association in the present study, particularly the IQ–mortality association (online supplemental material 1).

We then tested the moderating effect of father's social class. Adding the interaction term (age-11 IQ by father's social class) did not significantly improve the model fit versus the mutually adjusted model (mutually adjusted Akaike Information Criterion (AIC)=35 269.44, interaction AIC=35 270.08; $\chi^2\Delta$=1.36, p=0.24; interaction $R^2$=0.04). The interaction effect did not significantly predict all-cause mortality risk, with 95% CIs that included 1 (B=0.03, SE=0.02, HR=1.03, 95% CI 0.98 to 1.07, p=0.25). Adding the interaction term further introduced a multicollinearity issue (Variance Inflation Factor (VIF)=6.86)—strong collinearity between predictors (sex, father's social class, age-11 IQ and the interaction term) indicating some redundancy—which was not present in the mutually adjusted model (all VIFs<1.07).

Finally, we examined the association between age-11 IQ and all-cause mortality risk in each of the social classes (table 3). A 1 SD advantage in age-11 IQ significantly predicted moderately reduced mortality risk among those from unskilled (22% decrease), semiskilled (17% decrease) and skilled (17% decrease) backgrounds. The same direction of association, although not statistically significant and with wider 95% CIs that included 1, was observed among those from intermediate backgrounds. There was also no significant association between age-11 IQ and mortality risk among those from professional backgrounds, though this group was notably small (n=46). CIs for this group were much wider than those for other social

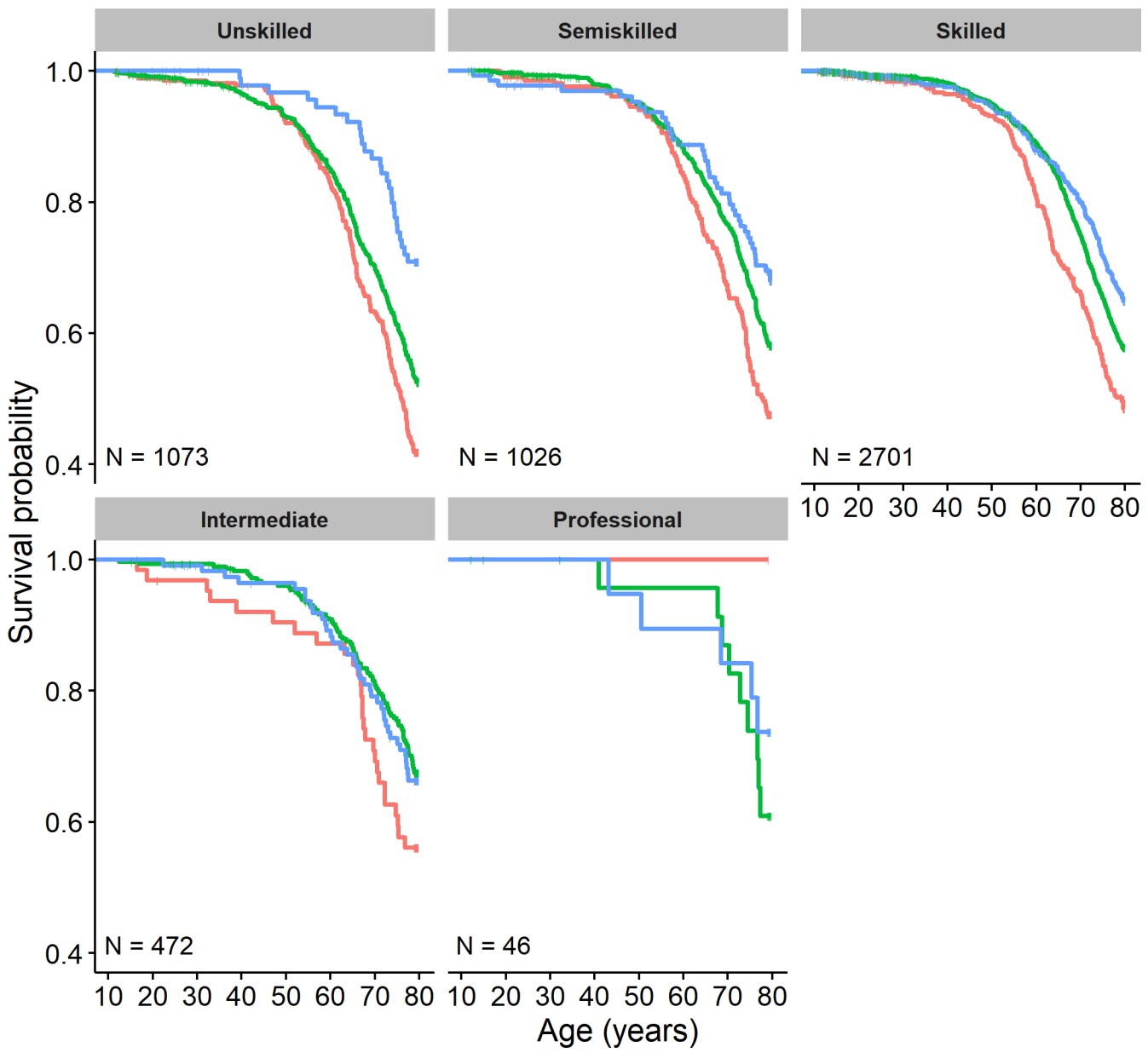

**Figure 1** Kaplan-Meier survival curves showing survival probability associated with IQ z-scores 1 SD below (red), around (green) or above (blue) the mean, split by father's occupational social class.

classes. Although not part of the preregistered analyses plan, at the advice of a reviewer, we examined IQ-mortality associations among those removed individuals with a missing social class (but complete other variables) as these individuals may represent a specific at-risk group. There was a marginally significant IQ–mortality association such that a 1 SD advantage in IQ was associated with a 27% decrease in mortality risk. Note, however, that this group was particularly variable—with wide CIs and a large SEs—reflecting its heterogeneity.

In further exploratory analyses suggested by reviewers, we observed no significant interaction between age-11 IQ and father's social class when not adjusting for sex or age at Scottish Mental Survey 1947 as confounders (online

supplemental material 1). We also observed no significant three-way interaction between IQ z-score, father's social class and sex (online supplemental material 1). Furthermore, a consistent pattern of results was observed when reanalysing the data using additive regression models (online supplemental material 1).

## DISCUSSION

The present study aimed to establish whether the association between higher childhood cognitive ability and reduced all-cause mortality risk is moderated by childhood socioeconomic circumstances. Whereas advantage in terms of childhood cognitive ability and father's social

**Table 2** Results from weighted Cox regression models predicting all-cause mortality risk

| Predictor | Univariate* | | | | Sex-adjusted and age-adjusted‡§ | | | | Mutually adjusted¶ | | | |
|---|---|---|---|---|---|---|---|---|---|---|---|---|
| | B | SE | HR (95% CI) | P value | B | SE | HR (95% CI) | P value | B | SE | HR (95% CI) | P value |
| IQ score (1 SD) | −0.21 | 0.02 | 0.81 (0.77 to 0.84) | <0.001 | −0.21 | 0.02 | 0.81 (0.78 to 0.85) | <0.001 | −0.19 | 0.02 | 0.82 (0.79 to 0.86) | <0.001 |
| Father's social class (one class) | −0.12 | 0.02 | 0.89 (0.85 to 0.93) | <0.001 | −0.12 | 0.02 | 0.89 (0.85 to 0.93) | <0.001 | −0.08 | 0.02 | 0.93 (0.88 to 0.97) | <0.001 |

Regression coefficients and HRs for IQ score and father's social class in univariate models, models adjusted for both IQ score and father's social class (and sex and age) and models adjusted for sex and age at Scottish Mental Survey 1947 and models adjusted for

*Univariate IQ model $R^2$=0.02.
†Univariate father's social class model $R^2$=0.01.
‡Sex and age-adjusted IQ model $R^2$=0.04.
§Sex and age-adjusted father's social class model $R^2$=0.02.
¶Mutually adjusted model $R^2$=0.04.

class independently predicted reduced risk of all-cause mortality, there was no evidence of a significant interaction effect in the whole-sample analysis. Although in the subgroup analysis we did observe a significant association between childhood cognitive ability–mortality risk in lower but not higher socioeconomic backgrounds, we did not explicitly compare associations between classes. Examination of the absolute difference in the risk of mortality likewise demonstrated that higher childhood cognitive ability and higher childhood socioeconomic circumstances significantly predicted fewer deaths per 10 000 person years, and that the interaction did not significantly predict additional deaths (online supplemental material 1). These results support previous observations that advantages in cognitive ability and socioeconomic circumstances have separate contributions to lower all-cause mortality risk,[6 7] and further suggest that the benefits of high cognitive ability apply regardless of social class.

The lack of a significant moderation effect by socioeconomic circumstances is contrary to our hypothesis and the few previous studies to examine this.[12 13] Notably, Jokela *et al*[13] observed a significant (although small) moderating effect of parental education, with a stronger cognitive ability–mortality association among those with more educated parents, rather than occupational social class as used here. Whereas parental education is thought to reflect the intellectual resources of the family, parental occupation is more indicative of the material resources of the family.[26] The benefits of higher cognitive ability for health, therefore, may not depend on the material resources available in childhood. Whereas Kajantie *et al*[12] did observe a significant moderation effect using parental occupational social class, only two categories were used (manual and middle class). Such a coarse measure of occupational social class may overestimate its contribution to all-cause mortality risk by shifting the estimates for each category towards the mean of included classes—downwards for lower social classes and upwards for higher social classes. Furthermore, it assumes no differences between social classes within each category; here, the mortality risk associated with skilled, intermediate and professional occupational classes was very different.

The main strength of the present study is the use of both socioeconomic circumstances and cognitive ability measured contemporaneously in childhood. In the present study, advantage in these factors independently predicted reduced mortality risk, with no evidence of multicollinearity or confounding by sex. Although we only included sex as a potential confounder, a more thorough investigation of bias using other potential confounders and mediators is available elsewhere,[9–11] including in the same Scottish Mental Survey 1947 subsamples.[6 7 14 15] Notably, childhood cognitive ability and childhood socioeconomic circumstances have been shown to predict later-life health, even after accounting for mediators such as education and socioeconomic circumstances in adulthood.[5 6 25 27] Measuring these factors contemporaneously avoids potential recall bias associated with retrospective

**Table 3** Regression coefficients and HRs for IQ score, adjusted for sex and age at Scottish Mental Survey 1947, in each of the Father's social classes

| | B | SE | HR (95% CI) | P value | R² | AIC |
|---|---|---|---|---|---|---|
| Unskilled (n=1073) | −0.25 | 0.05 | 0.78 [0.71 to 0.85) | <0.001 | 0.05 | 6668.66 |
| Semiskilled (n=1026) | −0.19 | 0.05 | 0.83 [0.75 to 0.91) | <0.001 | 0.03 | 5526.34 |
| Skilled (n=2701) | −0.19 | 0.03 | 0.83 [0.78 to 0.88) | <0.001 | 0.03 | 15 797.24 |
| Intermediate (n=472) | −0.13 | 0.08 | 0.88 [0.75 to 1.03) | 0.12 | 0.03 | 1873.17 |
| Professional (n=46) | 0.13 | 0.35 | 1.13 [0.57 to 2.26) | 0.72 | 0.05 | 112.63 |
| Missing social class (n=109)* | −0.32 | 0.17 | 0.73 [0.52 to 1.01) | 0.06 | 0.04 | 319.87 |

*Includes individuals with missing father's social class but with complete MHT scores and vitality status.
AIC, Akaike Information Criterion; MHT, Moray House Test.

measures of socioeconomic circumstances in particular.[28] Also, as individuals must survive until cognitive and socioeconomic measures are taken, examining these factors in childhood lessens potential selection effects (eg, lower cognitive ability in individuals who may die prior to adulthood). Furthermore, the present study examined all-cause mortality risk in a large sample over an entire life course (age 11–79 years old). These individuals have had longer exposure to mortality risk than the younger samples used in previous work.[12 13]

A limitation is that the sample is a product of the time and place in which they lived,[29] and this may limit the generalisability of the results. All individuals in the sample were born in 1936 and lived in Scotland in 1947. They survived the World War II and lived through the significant social, economic and educational changes that followed. Note that this also applies to previous work using the Helsinki Birth Cohort (born 1934–1944).[12] Childhood socioeconomic conditions (including occupational and educational opportunities of parents) in more modern samples, such as in the US National Longitudinal Survey of Youth,[13] are likely better overall than those captured in the present sample.[30] Indeed, relatively few individuals here originated from higher childhood social classes (intermediate or professional occupations). Furthermore, removal of individuals with missing childhood cognitive ability, childhood social class or vitality status likely resulted in a sample slightly biased towards higher cognitive ability. Note also that the sample selection analyses does not include those with no complete key variables (ie, all missing), who may constitute a very different population.

There are further limitations arising due to our use of father's occupational social class as a proxy measure of socioeconomic conditions. First, although deceased, absent and long-term unemployed fathers were coded according to their most recent occupation where possible, the removal of individuals with a missing father's occupational social class likely results in a sample that does not represent the full range of socioeconomic conditions, though this is not a limitation that is unique to this study.[12] Individuals raised in single-parent households have been shown to be at higher risk of mortality,[31] so removed

individuals may represent a particularly vulnerable population. Second, while father's occupational social class is useful for this age of cohort, it is perhaps less useful for younger cohorts for whom fathers are less likely to be the main breadwinners. More generally, the meaning of occupational social class is likely to have changed as new occupations are created and working conditions are improved[32]; thus, the nature of the role of social class may be different in more modern social classes. It is unclear whether other measures of childhood socioeconomic circumstances, such as parental education, play a role in moderating the benefits of higher childhood cognitive ability. Furthermore, these individuals had access to universal healthcare throughout their lives, unlike previous work using US-based samples,[13] likely reducing the importance of socioeconomic circumstances as both a predictor of mortality[33] and a moderator of the cognitive ability–mortality association. Finally, there are a number of unmeasured confounders, such as childhood health and earlier socioeconomic conditions, which may impact both childhood cognitive ability and mortality risk.[9] For example, socioeconomic disadvantage around birth (eg, lower family income and parental education) may lead to both poorer childhood cognitive ability[34] and increased mortality risk,[35] resulting in an arbitrary but strong association between higher cognitive ability and reduced mortality risk. However, previous work has shown that accounting for potential confounders such as earlier socioeconomic circumstances (including father's social class and quality of home environment) and childhood health results in only a small attenuation of the association between higher childhood cognitive ability and reduced mortality risk.[9] Furthermore, sensitivity analysis suggested that a particularly strong unmeasured confounder would be required to wholly account for the IQ–mortality association observed in the present study (online supplemental material 1).

Consistent with previous work, the present study demonstrates an association between higher cognitive ability and lower all-cause mortality risk. However, it also suggests that this association is not conditional on one's social class of origin. Whereas childhood social class was an important independent predictor of all-cause

mortality risk, there was no evidence that it moderated the role of childhood cognitive ability in predicting longevity. The benefits of higher childhood cognitive ability for lower all-cause mortality risk were observed across childhood social classes. This is an important consideration for public health policy and interventions: childhood cognitive ability is a source of health inequality independent from material socioeconomic resources in childhood and needs to be addressed separately. Future work should focus on building causal evidence and clarifying the mechanisms by which childhood cognitive ability and childhood socioeconomic conditions affect mortality risk.

**Author affiliations**
[1]Centre for Clinical Brain Sciences, The University of Edinburgh, Edinburgh, UK
[2]Department of Psychology, The University of Edinburgh, Edinburgh, UK
[3]Lothian Birth Cohorts, The University of Edinburgh, Edinburgh, UK

**Contributors** All authors contributed to the conceptualisation and design of this study. MHI contributed to data curation and formal analysis. MHI, DA and ID contributed to investigation; methodology; project administration; validation' visualisation; writing, original draft preparation; and writing, review and editing. ID contributed to funding acquisition and supervision.

**Funding** This work was supported by a Medical Research Council Mental Health Data Pathfinder award (MRC-MC_PC_17209) and by a UK cross-council Lifelong Health and Wellbeing Initiative (MRCG1001401), for which ID is the principal investigator. MHI, DA and ID are members of The University of Edinburgh Centre for Cognitive Ageing and Cognitive Epidemiology, part of the cross-council Lifelong Health and Wellbeing Initiative (MR/K026992/1).

**Competing interests** None declared.

**Patient consent for publication** Not required.

**Ethics approval** This submission is covered by the ethics application 'Lifelong health and wellbeing of the 'Scotland in Miniature' cohort: the 6-Day Sample of the Scottish Mental Survey' (REC# 12/SS/0024, Scotland A Research Ethics Committee). The study was approved by the local ethics committee and conformed to the Declaration of Helsinki.

**Provenance and peer review** Not commissioned; externally peer reviewed.

**Data availability statement** Data are available upon reasonable request. Historical data from the Scottish Mental Survey 1947 are available upon request from the Department of Psychology, The University of Edinburgh. Access is contingent on local governance approvals. Routinely collected records, including national death records, are available and linkable through NHS Information Services Division and the eData Research and Innovation Service (https://www.isdscotland.org/Products-and-Services/EDRIS/). Access is contingent on ethical and Public Benefit and Privacy Panel approvals.

**ORCID iD**
Matthew Henry Iveson http://orcid.org/0000-0002-7242-0456

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
