## [Reviewer comments · BMJ Open]

ARTICLE DETAILS

TITLE (PROVISIONAL)	Do childhood socioeconomic circumstances moderate the association between childhood cognitive ability and all-cause mortality across the life course? Prospective observational study of the 36-Day Sample of the Scottish Mental Survey 1947.
AUTHORS	Iveson, Matthew; Altschul, Drew; Deary, Ian

VERSION 1 – REVIEW

REVIEWER	Stephen Gilman in collaboration with Jing Yu and Reeya Patel Eunice Kennedy Shriver National Institute of Child Health and Human Development
REVIEW RETURNED	29-May-2020

GENERAL COMMENTS	Iveson and colleagues used data from the Scottish Mental Survey 1947, a large population-based sample of 5,318 individuals born in 1936 who were administered intelligence tests in 1947 and whose vital status was ascertained through 2015, to address the question of whether or not cognitive ability at 11 years is a stronger or weaker predictor of mortality risk depending on children's socioeconomic circumstances – here measured by father's occupation. They found that it was not: that the association between children's cognitive performance was protective against mortality for all children irrespective of their father's occupation. The conclusion from their study is that, as the manuscript puts it, "the benefits of high cognitive ability apply regardless of social class." 1. The introduction is framed entirely in terms of analytic models – the mediation, moderating, and confounding influences of socioeconomic circumstances in the association between cognitive ability and mortality. This framing essentially leaves unclear why these difference models may be operating and as a consequence what the motivation is for investigating potential moderating effects and what the fundamental basis of the study is. What is the overarching causal model motivating the study, and what would be the implications of evidence for or against moderation? For example, is it that cognitive performance – either as an indicator of cognitive reserve or of resilience capacity – offers greater health benefits in the context of relative affluence than in relative deprivation? Why would this be the case? Relatedly, there is strong evidence that childhood socioeconomic circumstances influence children's neurocognitive development; how is this process taken into account (including the potential mediating role of cognitive abilities in the association between childhood socioeconomic circumstances and mortality)?
---

	2. The Scottish Mental Survey 1947 provides a unique opportunity to address the issue of childhood cognition and long-term mortality given its population-based sampling, large sample size, and long-term follow-up. To what extent, however, might the findings be specific to that period in time or place? The manuscript addresses this issue very nicely on page 16, particularly noting the small number of individuals in the sample in the highest social class. What would be most helpful is to review the results of comparable studies also in terms of their time and place. As the social class structure changes over time (perhaps as income inequality increases or decreases), would this influence at the individual level the contribution of cognitive performance to long-term health? 3. There are several methodologic issues. a) Analyzing social class as a continuous variable imposes a strong assumption that the moderating effect is linear (on the hazards scale); the manuscript indicates that this was done because otherwise the proportional hazards assumption is violated. As a result, the current analyses average over potentially significant non-linear effects of the class*cognition interaction and non-proportional effects over time. Rather than obscuring these processes, the analyses should explore them. Based on the current analyses, I would not yet conclude that moderation does not exist. b) The risk set included in the analyses ranged from birth through death; however, children were not at risk for mortality as a function of their age 11 cognition until they were 11 years old. It would seem more accurate to model survival time ranging from 11 to death/censoring. c) Could moderating effects vary by sex? There is evidence to suggest that they might. 4. The manuscript concludes that childhood cognitive ability is a source of health inequality that needs to be addressed separately from socioeconomic status. I'm not sure this is a fair representation of the data or the broader literature. Given that socioeconomic circumstances in childhood have a strong influence on children's development including neurodevelopment, I would argue that socioeconomic circumstances are the major source of health inequality. When the authors refer to addressing various sources of health inequality, what types of interventions do they have in mind?
--	--

REVIEWER	Else Foverskov University of Copenhagen, Denmark
REVIEW RETURNED	16-Jun-2020

GENERAL COMMENTS	This paper examines if childhood social class moderates an association between childhood IQ and mortality up until age 79. Overall, the manuscript is well written and the methods are well laid out. I am missing considerations of interactions on additive vs. multiplicative scale as well as consideration of lack of statistical power to detect an interaction. I think the results could be strengthened by including additional sensitivity analyses. Specify that you are using cox models in the method section of the abstract. The introduction could benefit from including an explanation of the reasons why childhood social class may potentially moderate an association between childhood IQ and mortality.
---

	Is the 36-day sample of the Scottish Mental Survey (the 6291 individuals) representative of the background population? You use both IQ and social class as numeric exposures. Did you examine if there was a linear relationship between these variables and the log hazard? In the descriptive part of the analyses, you use the original MHT score and then you switch to using the converted IQ-type score in the survival analyses. Is there any reasons for not using the IQ score in all of the analyses? In the graph for childhood social class = professional (Figure 1) is the flat red line (IQ z-score 1 or more SD below) due to there not being any people in this group? Table 3, which shows regression coefficients for IQ scores in each of the social classes, is presented as a way to better assess interaction effects, but there is no explanation of how this analysis can be used to make conclusions about an interaction effect. Also, in the results section the results from the analysis are never directly interpreted with reference to what it tells us about an interaction effect. Interaction analyses are notorious for requiring very large samples and I wonder if low statistical power is a limitation worth discussing. In this regard, I think it would be interesting to see the results from an analysis where the measure of childhood social class is collapsed into two or three groups. I recognize the argument that “a coarse measure of occupational social class may overestimate its contribution”, but on the other hand I am not sure it is reasonable to think that associations can be detected in social class groups only including 472 and 46 people. There is arguably broad agreement that it is more relevant from a public health perspective to assess additive interactions instead of multiplicative interactions, and I think the manuscript would be strengthened by including an assessment of interaction on the additive scale. This could be done by calculating relative excess risk due to interaction (RERI) or as an alternative using an additive hazards model to estimate absolute deviation from additivity. - Li R, Chambless L. Test for Additive Interaction in Proportional Hazards Models. Ann Epidemiol. 2007;17(3):227-236. - Rod NH, Lange T, Andersen I, Marott JL, Diderichsen F. Additive interaction in survival analysis: use of the additive hazards model. Epidemiology. 2012;23(5):733-737. How many people are excluded due to their parents not having a job back in 1947? Is it all 130 people with missing childhood social class information? Assessing the association between childhood IQ and mortality in this group may give an indication of the implications of not including them in the analysis.
--	---

REVIEWER	Gitte Lindved Petersen Translational Type 1 Diabetes Research, Steno Diabetes Center Copenhagen & Section of Epidemiology, Department of Public Health, University of Copenhagen, Denmark
REVIEW RETURNED	21-Jun-2020

GENERAL COMMENTS	This longitudinal cohort study assesses the potential interaction between childhood cognitive function and socioeconomic position in relation to all-cause mortality. The study sample includes 5,328 individuals born in 1936 and living in Scotland in 1947 who were followed prospectively until death or end of study follow-up at age 79 years in 2015. The paper appears overall clearly and well written. However, I have a few major concerns that I hope the authors will address and a number of minor suggestions. 1. Is the research question or study objective clearly defined? The authors refer to previous studies treating socioeconomic circumstances as either mediators or confounders in the relationship between cognitive function and mortality (p. 4, l. 15-26). As also indicated by the authors, this relies on different hypotheses than socioeconomic circumstances moderating the potential effect of cognitive function on mortality. In the introduction I miss a hypothesis behind a potential interaction, i.e. why would one expect cognitive function in childhood to influence mortality differently in some strata of childhood SEP compared to others? In relation to above, I also miss a clearer argumentation for conducting this study besides 'shining a light in a dark place'. P. 4-5, l. 58-10: The authors mention that previous research is limited by only including men, by short follow-up time and by using a measure of cognitive ability from young adulthood rather than childhood. I believe it would build a stronger argument for conducting the present study if the authors elaborated a little on how this compromise the results of the previous studies. 3. Is the study design appropriate to answer the research question? Not much attention is given to confounding. Sex-adjusted results are presented in Table 2, but the rationale behind this adjustment should be included in the paper. How did the authors identify potential confounders, and have the authors considered potential unmeasured confounders? One could for example speculate that early life exposures (for example growth or health/disease) represent confounders that can influence cognitive function as well as risk of early death. 4. Are the methods described sufficiently to allow the study to be repeated? It is not clear how the authors reached the study sample of 5,318 from the original sample including 6,291 individuals (excluding 575+130+13=718). P. 6, l. 40: The description of the MHT should be more specific. Please rephrase the sentence ending with '...and so on.' To avoid confusion, please avoid referring to 'odds' in relation to results from Cox regression models (p. 7, l. 52-53).
---

P. 7, l. 5-7: I find it unclear how father's occupational class was identified. E.g. how far back in time could a given 'most recent occupation' be to be accepted? And how was 'without occupation' and 'long-term unemployed' defined?

7. If statistics are used are they appropriate and described fully?

P. 14, l. 10-21: To my knowledge, adding an interaction term to a model ($x_1 \times x_2$) will naturally result in some level of structural multicollinearity because the main variables (x_1 and x_2) already exist in the model. It would be helpful if the authors assessed and discussed whether the detected multicollinearity is purely a structural issue in the interaction model or if it is present in the data itself (if the purpose of the analysis is to address a causal question rather than predicting).

10. Are they presented clearly?

P. 11, l. 22-34: Instead of solely reporting a p-value, please report the absolute and relative differences between subgroups as this is much more informative to the reader.

Table 1:

- Please add range rather than max for the MHT score. To help the reader, I suggest adding that the MHT score is measuring intelligence.

Table 2:

- It should be added that the table presents results from Cox regression models and that the outcome is all-cause mortality.
- Main effects are not meaningful when the interaction term is included in the model. I suggest removing the last column from this table (including the corresponding text p. 14, l. 17-21) and instead reporting the results for the interaction term in text relating to Table 3 as the interaction is further addressed there.

11. Are the discussion and conclusions justified by the results

Several pieces have been published about the pitfalls of relying too strongly on statistical significance alone (see for example Greenland et al. 2016 'Statistical tests, P values, confidence intervals, and power: a guide to misinterpretations' in European Journal of Epidemiology). In my opinion, the authors in general put too much weight on statistical significance when interpreting the results from their statistical analyses. Instead the results should be interpreted bearing in mind hypotheses, study design, effect sizes, confidence limits, sample size, the potential influence of bias etc.

12. Are the study limitations discussed adequately?

It should be discussed if childhood cognitive function and childhood socioeconomic position could suffer from multicollinearity and how this may affect the interpretation of the results.

Paternal occupational class is used as a proxy of childhood SEP. Although this is a well-recognized indicator of childhood SEP, it

	holds strengths and disadvantages that should be discussed a little further:  • Back in time the breadwinner of the family was most likely to be the father and thus for the majority of the population father's occupational class has been a reasonably good indicator of childhood SEP. But how does it influence the results that it is not possible to categorize for example children of single mothers? • How do the authors expect the limitations (such as discussed p. 16, l. 26-42) of this measure to influence their findings? P. 11, l. 3-19: From the comparison of excluded and included individuals the authors conclude that only a minor difference in cognitive performance exists and that there is no difference in social class. I question this interpretation to be valid for the full sample. As far as I understand from the methods section the exclusions were due to missing cognitive performance score, or social class, or both. Thus, the reported numbers must be based on those with available information on one of the two variables ignoring those with missing information on both. I question that individuals with missing information on both (or all) key variables are comparable to those that are only missing information on one. P. 15, l. 40-42: Please elaborate on how a dichotomization of social class may overestimate its effect on mortality. P. 16, l. 12: The authors state that the study is based on a nationally representative sample. I question this as I believe a selection into the study exists. The 'complete cases' approach (i.e. including only individuals with full information on all key variables) excludes a number of study participants who did for example not complete the cognitive test or did not have information on paternal occupational class. Moreover, paternal occupational class was coded missing if the father was 'without occupation' or 'unemployed', and finally, it is not clear how children of single mothers were handled. I find it reasonable to argue that children of unemployed fathers and single mothers must represent a socially vulnerable group, especially back in time when this cohort was born. A fairly large number of individuals were excluded due to missing information on childhood cognitive ability. What was the reasons for this? To the reader this is important information in order to evaluate the probability of this being completely random or not. I am not convinced that the excluded individuals are comparable to those included, nor to the population in general. I believe that the paper can be markedly improved by a more thorough analyses of the potential selection bias affecting the results. This could be done by means of inverse probability weighing or at least a more thorough analysis of the potential bias in order to quantify the potential impact on the findings. P. 16, l. 19-42: I agree with the authors that a limitation of life course studies is that the generalizability to younger generations can be
--	---

	compromised by societal changes across time. Also, a specific (categorization of a) measure of occupational class may not suit data from different time periods equally good, as the authors describe. However, I find it unclear whether the authors merely suggest that studies of younger cohorts need to use other measures of occupational class to ensure sensitivity to detect social inequality or if they believe that social class will play no (or only a minor) role in the relationship between cognitive function and mortality in younger cohorts. Additional comments To improve readability of the paper, each key concept (i.e. socioeconomic position and childhood cognitive ability) should be given one name based on a conceptually sound reflection and these names should be used consistently throughout the paper. Number of decimals and ways of reporting p-values should be reasonable and consistent throughout the paper.
--	--

VERSION 1 – AUTHOR RESPONSE

Reviewer 1

Iveson and colleagues used data from the Scottish Mental Survey 1947, a large population-based sample of 5,318 individuals born in 1936 who were administered intelligence tests in 1947 and whose vital status was ascertained through 2015, to address the question of whether or not cognitive ability at 11 years is a stronger or weaker predictor of mortality risk depending on children’s socioeconomic circumstances – here measured by father’s occupation. They found that it was not: that the association between children’s cognitive performance was protective against mortality for all children irrespective of their father’s occupation. The conclusion from their study is that, as the manuscript puts it, “the benefits of high cognitive ability apply regardless of social class.”

1. The introduction is framed entirely in terms of analytic models – the mediation, moderating, and confounding influences of socioeconomic circumstances in the association between cognitive ability and mortality. This framing essentially leaves unclear why these difference models may be operating and as a consequence what the motivation is for investigating potential moderating effects and what the fundamental basis of the study is. What is the overarching causal model motivating the study, and what would be the implications of evidence for or against moderation? For example, is it that cognitive performance – either as an indicator of cognitive reserve or of resilience capacity – offers greater health benefits in the context of relative affluence than in relative deprivation? Why would this be the case? Relatedly, there is strong evidence that childhood socioeconomic circumstances influence children’s neurocognitive development; how is this process taken into account (including the potential mediating role of cognitive abilities in the association between childhood socioeconomic circumstances and mortality)?

RESPONSE: We agree with the reviewer that the mechanisms for the moderation effect and the rationale of the study were unclear. We have developed the discussion of previously-asserted mechanisms, including that better socioeconomic conditions enable higher cognitive ability to benefit longevity through access and resources for healthier behaviours. Note, however, that we attempt to be reasonably agnostic about the nature of any moderation as we do not have the variables necessary to directly test different mechanisms. [Page 4, paragraph 2]

We agree that childhood socioeconomic circumstances have important implications for neurodevelopment. Here, childhood cognitive ability and social class were measured at the same time (age 11); no earlier measure of childhood socioeconomic circumstances was used. However, the potential mediating role of early-life socioeconomic circumstances has been explored by a variety of previous studies, including in the same sample and subsample as used here. The majority of these studies find that early-life socioeconomic circumstances do not confound the association between early-life cognitive ability and mortality, even when conservatively accounting for all shared variance within a household. We have included discussion of this in the Introduction. [Page 4, paragraph 1; Page 5, paragraph 2].

2. The Scottish Mental Survey 1947 provides a unique opportunity to address the issue of childhood cognition and long-term mortality given its population-based sampling, large sample size, and long-term follow-up. To what extent, however, might the findings be specific to that period in time or place? The manuscript addresses this issue very nicely on page 16, particularly noting the small number of individuals in the sample in the highest social class. What would be most helpful is to review the results of comparable studies also in terms of their time and place. As the social class structure changes over time (perhaps as income inequality increases or decreases), would this influence at the individual level the contribution of cognitive performance to long-term health?

RESPONSE: As the reviewer points out, we discuss the limitations linked to this specific sample in the Discussion. We have developed this to include reference to the samples of other similar studies, one of which is of similar age (and so subject to similar period/place considerations) and one of which is slightly younger. [Page 18, paragraph 2]

3. There are several methodologic issues. a) Analyzing social class as a continuous variable imposes a strong assumption that the moderating effect is linear (on the hazards scale); the manuscript indicates that this was done because otherwise the proportional hazards assumption is violated. As a result, the current analyses average over potentially significant non-linear effects of the class*cognition interaction and non-proportional effects over time. Rather than obscuring these processes, the analyses should explore them. Based on the current analyses, I would not yet conclude that moderation does not exist. b) The risk set included in the analyses ranged from birth through death; however, children were not at risk for mortality as a function of their age 11 cognition until they were 11 years old. It would seem more accurate to model survival time ranging from 11 to death/censoring. c) Could moderating effects vary by sex? There is evidence to suggest that they might.

RESPONSE: We agree that using a continuous version of social class makes the assumption that the moderating effect is linear. As the reviewer notes, this was done to avoid violating the proportional hazards assumption and because the numeric variable provided similar fit. We do conduct a planned examination of the IQ-mortality association within each social class and find significant associations (of similar magnitude) within the Unskilled, Semi-skilled and Skilled classes. Although we find non-significant associations in the Intermediate and Professional groups we attribute this to the relatively low size of each group, noting the wider confidence intervals that span the null. [Page 15, paragraph 2]

We agree that children necessarily had to be alive at the time of their cognitive ability assessment to be included in the sample. We have updated all analyses to reflect survival between 4th June 1947 and 30th November 2015. [Page 8, paragraph 1]

Although there is evidence for a moderating effect that varies by sex we did not set out to test this and the 3-way interaction was not part of the pre-registered analyses. Additionally, we do not make any predictions regarding this interaction. As such, we have included a test of the 3-way interaction in the Supplementary Material and have labelled it as exploratory, summarising it in the Results section of the main text. Note, however, that there was no significant 3-way interaction ($B = 0.003$, $p = 0.94$) with

CIs that span 1, likely due to the increased demand for statistical power required by a complex interaction. [Supplementary Material]

4. The manuscript concludes that childhood cognitive ability is a source of health inequality that needs to be addressed separately from socioeconomic status. I'm not sure this is a fair representation of the data or the broader literature. Given that socioeconomic circumstances in childhood have a strong influence on children's development including neurodevelopment, I would argue that socioeconomic circumstances are the major source of health inequality. When the authors refer to addressing various sources of health inequality, what types of interventions do they have in mind?

RESPONSE: We do not intend to suggest that cognitive ability is any more or less an important consideration than socioeconomic circumstances when addressing health inequality. We conclude that they are independent, with the caveats that we use only one proxy for socioeconomic conditions and that this measure brings limitations. We have improved the discussion of these limitations and their consequences in the Discussion, including noting that earlier socioeconomic conditions may confound the IQ-mortality association. [Page 19, paragraph 1]

Reviewer 2

This paper examines if childhood social class moderates an association between childhood IQ and mortality up until age 79. Overall, the manuscript is well written and the methods are well laid out. I am missing considerations of interactions on additive vs. multiplicative scale as well as consideration of lack of statistical power to detect an interaction. I think the results could be strengthened by including additional sensitivity analyses.

Specify that you are using cox models in the method section of the abstract.

RESPONSE: We have included this in the abstract as requested. [Page 2]

The introduction could benefit from including an explanation of the reasons why childhood social class may potentially moderate an association between childhood IQ and mortality.

RESPONSE: We agree that the mechanisms underlying the potential moderation effect and the resulting rationale of the study were unclear. We have developed the discussion of previously-asserted mechanisms, including that better socioeconomic conditions enable higher cognitive ability to benefit longevity through access and resources for healthier behaviours. Note, however, that we attempt to be reasonably agnostic about the nature of any moderation as we do not have the variables necessary to directly test different mechanisms. [Page 5, paragraph 1]

Is the 36-day sample of the Scottish Mental Survey (the 6291 individuals) representative of the background population?

RESPONSE: The 36-day sample of the Scottish Mental Survey 1947 consists of individuals born on one of 36 birth days throughout 1936. In this sense, the 36-day sample is a randomised sample of the larger population. Their representativeness in terms of childhood cognitive ability and socioeconomic factors (family size, overcrowding, etc.) has been demonstrated in previous work (The Scottish Council for Research in Education, 1949). We have noted this in the Methods section. [Page 6, paragraph 2]

You use both IQ and social class as numeric exposures. Did you examine if there was a linear relationship between these variables and the log hazard?

RESPONSE: We detail correlations between survival time and both MHT scores and father's social class (as continuous variables) in the Results section, noting small positive correlations in each case. [Page 9, paragraph 4] Furthermore, the linear relationship of each variable with the log hazard is examined in the univariate Cox models (Table 2), in which we note a negative linear association with both predictors. [Page 14]

We believe, however, that the reviewer is referring to the fact that using a continuous version of social class makes the assumption that the moderating effect is linear. We note in the Analysis that this was done to avoid violating the proportional hazards assumption and because the numeric variable provided similar fit. [Page 9, paragraph 2] The linear association between the interaction and the log hazard is examined in the Results section [Page 15, paragraph 1]. Although we do not examine the non-linear interaction effect directly (again, to meet the proportional hazards assumption) we do examine the linear associations between continuous IQ and mortality hazard in each social class. [Page 15, paragraph 2]

In the descriptive part of the analyses, you use the original MHT score and then you switch to using the converted IQ-type score in the survival analyses. Is there any reasons for not using the IQ score in all of the analyses?

RESPONSE: The descriptives use raw MHT scores as they are the most comparable measure of cognitive ability between the full and analytic samples. Scores are then transformed into IQ-type scores to aid modelling and interpretation in the survival analyses.

In the graph for childhood social class = professional (Figure 1) is the flat red line (IQ z-score 1 or more SD below) due to there not being any people in this group?

RESPONSE: The flat red line indicates that all individuals with an IQ z-score 1 or more SDs below the mean survive until the censor time (i.e., there is no mortality hazard). However, as we note in the Results section, this sub-group is small and so this trend represents only a few individuals. [Page 15, paragraph 2]

Table 3, which shows regression coefficients for IQ scores in each of the social classes, is presented as a way to better assess interaction effects, but there is no explanation of how this analysis can be used to make conclusions about an interaction effect. Also, in the results section the results from the analysis are never directly interpreted with reference to what it tells us about an interaction effect.

RESPONSE: In the pre-registration we specified that we would run a within-social class examination of the IQ-mortality association as a follow-up to the hypothesised significant interaction effect. In particular, we hypothesised that a stronger IQ-mortality association would be observed in the higher social classes. Although no significant interaction was observed we conducted the within-social class analysis as planned. This stage of the analyses leads to an important note about the sample size of higher social classes [Page 15, paragraph 2] and discussion of the implications. [Page 18, paragraph 2]

Interaction analyses are notorious for requiring very large samples and I wonder if low statistical power is a limitation worth discussing. In this regard, I think it would be interesting to see the results from an analysis where the measure of childhood social class is collapsed into two or three groups. I recognize the argument that “a coarse measure of occupational social class may overestimate its contribution”, but on the other hand I am not sure it is reasonable to think that associations can be detected in social class groups only including 472 and 46 people.

RESPONSE: We agree that interactions require substantial sample sizes for adequate statistical power. Notably, the sample size here (N = 5318) is comparable to previous work (e.g., Kajantie et al, 2010 N = 2786). Note also that the interaction was assessed on the linear interaction effect and so should be less affected by the small number of Professional individuals.

In regards to the collapsed analysis, we did not plan this as part of the pre-registration. As the reviewer notes, we consider a binary measure of social class to be too crude and to be at risk of overestimating the interaction effect. Furthermore, the decision on which groups to collapse is problematic. While the Intermediate and Professional groups are smallest, they are also very different in terms of background descriptives (Table 1), including a mean difference in Moray House Test score of 9 points. We have therefore decided not to conduct a re-analysis based on a combined social class variable.

There is arguably broad agreement that it is more relevant from a public health perspective to assess additive interactions instead of multiplicative interactions, and I think the manuscript would be strengthened by including an assessment of interaction on the additive scale. This could be done by calculating relative excess risk due to interaction (RERI) or as an alternative using an additive hazards model to estimate absolute deviation from additivity.

- Li R, Chambless L. Test for Additive Interaction in Proportional Hazards Models. *Ann Epidemiol.* 2007;17(3):227-236.

- Rod NH, Lange T, Andersen I, Marott JL, Diderichsen F. Additive interaction in survival analysis: use of the additive hazards model. *Epidemiology.* 2012;23(5):733-737.

RESPONSE: We appreciate the suggestion of additive hazard models and the links to appropriate references. In the main text, we conduct Cox proportional hazard regression in keeping with previous work. Furthermore, the analysis is conducted according to the pre-registered analysis plan. We appreciate that additive models allow the effects of covariates – including interaction effects – to vary over time. For example, advantage in childhood cognitive ability may benefit survival in particular socioeconomic groups but only in early life. We also appreciate that additive models help to provide better information about the effect in the context of the underlying hazard – a small hazard ratio for the interaction may still be important if the underlying hazard is large. However, as an additive interaction effect was not planned or hypothesised we have included the re-analysis using additive models in the Supplementary Material, summarising it at the end of the Results section of the main text. The re-analysis is very much in keeping with the proportional hazards approach used in the main text. Examination of the IQ-mortality association within classes suggests that there is a stronger IQ-related reduction in deaths per person year among those in lower social classes (following the same trend as in the main text), though the interaction effect itself was only marginally significant. We therefore treat the interpretation with caution. [Supplementary Material]

How many people are excluded due to their parents not having a job back in 1947? Is it all 130 people with missing childhood social class information? Assessing the association between childhood IQ and mortality in this group may give an indication of the implications of not including them in the analysis.

RESPONSE: As part of the recording of father's occupation in 1947, unemployed, absent or deceased fathers were recorded as their most recent occupation where possible. We have better described this process in the Methods section. [Page 7, paragraph 2]

We agree with the reviewer that those removed from the sample, particularly those with missing father's occupational social class are an interesting group. We have included an analysis of the IQ-mortality association in those with 'missing' childhood social class in the Results section, and have noted that it was not part of the pre-registered analyses. This showed a marginally-significant negative association of stronger magnitude than in any other social class group; however, caution is advised due to the exploratory nature of this analysis and the particularly wide confidence intervals and large standard errors. [Page 15, paragraph 2; Page 16].

We have also added discussion of the implications of removing those with missing variables, including the possibility of selection bias, in the Discussion. [Page 18, paragraph 3]

Reviewer 3

This longitudinal cohort study assesses the potential interaction between childhood cognitive function and socioeconomic position in relation to all-cause mortality. The study sample includes 5,328 individuals born in 1936 and living in Scotland in 1947 who were followed prospectively until death or end of study follow-up at age 79 years in 2015.

The paper appears overall clearly and well written. However, I have a few major concerns that I hope the authors will address and a number of minor suggestions.

1. Is the research question or study objective clearly defined?

The authors refer to previous studies treating socioeconomic circumstances as either mediators or confounders in the relationship between cognitive function and mortality (p. 4, l. 15-26). As also

indicated by the authors, this relies on different hypotheses than socioeconomic circumstances moderating the potential effect of cognitive function on mortality. In the introduction I miss a hypothesis behind a potential interaction, i.e. why would one expect cognitive function in childhood to influence mortality differently in some strata of childhood SEP compared to others?

RESPONSE: We thank the reviewer for their comment and accept that the hypotheses were unclear. We have added further discussion of the hypothesised mechanisms to the Introduction. [Page 4, paragraph 2]

In relation to above, I also miss a clearer argumentation for conducting this study besides 'shining a light in a dark place'.

RESPONSE: We agree that the set-up for the study was not sufficiently clear. We have improved the rationale for the study in the Introduction by discussing the limitations of previous work, the advantages of childhood cognitive ability, and the need for further assessments of moderating effects. [Page 5, paragraph 1]

P. 4-5, l. 58-10: The authors mention that previous research is limited by only including men, by short follow-up time and by using a measure of cognitive ability from young adulthood rather than childhood. I believe it would build a stronger argument for conducting the present study if the authors elaborated a little on how this compromise the results of the previous studies.

RESPONSE: We agree that these limitations are worth expanding upon. We have added discussion of these prior limitations into the Introduction. [Page 5, paragraph 1]

3. Is the study design appropriate to answer the research question?

Not much attention is given to confounding. Sex-adjusted results are presented in Table 2, but the rationale behind this adjustment should be included in the paper. How did the authors identify potential confounders, and have the authors considered potential unmeasured confounders? One could for example speculate that early life exposures (for example growth or health/disease) represent confounders that can influence cognitive function as well as risk of early death.

RESPONSE: We adjusted for sex in the analyses due to the well-established association between sex and mortality risk, and to be consistent with previous studies examining moderation effects (Jokela et al., 2009). We have clarified this in the Methods section. [Page 8, paragraph 2]

As our research question focussed on the association between mortality risk and both childhood IQ and socioeconomic circumstances we opted not to include other covariates such as education or adult social class. Furthermore, fuller investigations of confounding (but not moderation) in the IQ-mortality association have been made as part of previous work, including in the same sample (e.g., Calvin et al., 2011; Calvin et al., 2017; Iveson et al., 2018). As the reviewer suggests, there a number of other covariates that play a role in confounding/mediating the relationship between IQ and mortality, but previous work has generally shown IQ-mortality associations remain after adjustment. We have noted in the Discussion that a more detailed exploration of confounders in the same sample can be found elsewhere. [Page 17, paragraph 3]

As with other studies, a number of unmeasured factors could have acted as confounders. We have acknowledged this, pointing to childhood health as a particular source of confounding, in the limitations section of the Discussion. [Page 19, paragraph 1]

4. Are the methods described sufficiently to allow the study to be repeated?

It is not clear how the authors reached the study sample of 5,318 from the original sample including 6,291 individuals (excluding 575+130+13=718).

RESPONSE: Although we specified the number of individuals removed due to missing date of death (but recorded as deceased) we did not specify the number of individuals removed due to missing vitality status. We have added this so that the sentence now reads:

"To create the analytic sample (N = 5318), we removed individuals with missing childhood cognitive ability (N = 575), missing childhood socioeconomic circumstances (N = 130), missing vitality status (N = 312) or missing date of death (if deceased, N = 12). Note that these individuals could have more than one key variable missing." [Page 6, paragraph 3]

P. 6, l. 40: The description of the MHT should be more specific. Please rephrase the sentence ending with '...and so on.'

RESPONSE: We have provided more information about the MHT and its administration in the Measures section. Furthermore, the sentence about test items has been rephrased to read: “The MHT consists of 71 items including: verbal reasoning, word classification, mixed sentences, same-opposites, analogies, proverbs, arithmetic, cypher decoding, following directions, proverbs, spatial problem-solving and practical items.” [Page 7, paragraph 1]

To avoid confusion, please avoid referring to ‘odds’ in relation to results from Cox regression models (p. 7, l. 52-53).

RESPONSE: We have amended this to read “to assess the association between standardised IQ scores and reversed father’s occupational social class (separately) and mortality risk.” [Page 8, paragraph 3]

P. 7, l. 5-7: I find it unclear how father’s occupational class was identified. E.g. how far back in time could a given ‘most recent occupation’ be to be accepted? And how was ‘without occupation’ and ‘long-term unemployed’ defined?

RESPONSE: Father’s occupational social class was recorded in 1947 as part of the Sociological Schedule, and was primarily completed by school officials (head teacher or medical staff). Schedule completers were given detailed guidance on how to record occupations. Those without occupations, those long-term unemployed, and those deceased/absent were recorded as their most recent occupation where possible but were otherwise recorded as missing. This has been clarified in the description of the measure. [Page 7, paragraph 2]

7. If statistics are used are they appropriate and described fully?

P. 14, l. 10-21: To my knowledge, adding an interaction term to a model ($x_1 \times x_2$) will naturally result in some level of structural multicollinearity because the main variables (x_1 and x_2) already exist in the model. It would be helpful if the authors assessed and discussed whether the detected multicollinearity is purely a structural issue in the interaction model or if it is present in the data itself (if the purpose of the analysis is to address a causal question rather than predicting).

RESPONSE: The multicollinearity issue was introduced by adding the interaction term, so we agree that this is likely a structural issue in the interaction model. We have clarified this in the text, including the VIFs of the mutually-adjusted model for comparison. [Page 15, paragraph 1]

10. Are they presented clearly?

P. 11, l. 22-34: Instead of solely reporting a p-value, please report the absolute and relative differences between subgroups as this is much more informative to the reader.

RESPONSE: We have briefly highlighted differences between Professional and Unskilled subgroups in the text as suggested. [Page 9, paragraph 4]

Table 1:

- Please add range rather than max for the MHT score. To help the reader, I suggest adding that the MHT score is measuring intelligence.

RESPONSE: The max was included to indicate the maximum possible MHT score (rather than the highest observed MHT score). We have replaced this with the observed range of MHT scores as suggested. [Page 11]

Table 2:

- It should be added that the table presents results from Cox regression models and that the outcome is all-cause mortality.

RESPONSE: We have amended the title of Table 2 to read:

“Table 2. Results from weighted Cox regression models predicting all-cause mortality risk. Regression coefficients and hazard ratios (HR) for IQ score and Father’s social class in univariate models, models adjusted for sex and age at Scottish Mental Survey 1947 and, models adjusted for both IQ score and Father’s social class (and sex and age), and a model including an interaction term (and sex).” [Page 14]

- Main effects are not meaningful when the interaction term is included in the model. I suggest removing the last column from this table (including the corresponding text p. 14, l. 17-21) and instead reporting the results for the interaction term in text relating to Table 3 as the interaction is further addressed there.

RESPONSE: We thank the reviewer for their suggested clarifications. While we believe that main effects can be meaningful in the presence of an interaction (e.g., where there is an IQ effects across all SES levels) we agree that it isn't the case here and that Table 2 may be misinterpreted. As suggested, we have removed the 'interaction model' column in Table 2 and have instead moved inferential statistics for the interaction effect in-text. [Page 15, paragraph 1]

11. Are the discussion and conclusions justified by the results

Several pieces have been published about the pitfalls of relying too strongly on statistical significance alone (see for example Greenland et al. 2016 'Statistical tests, P values, confidence intervals, and power: a guide to misinterpretations' in European Journal of Epidemiology). In my opinion, the authors in general put too much weight on statistical significance when interpreting the results from their statistical analyses. Instead the results should be interpreted bearing in mind hypotheses, study design, effect sizes, confidence limits, sample size, the potential influence of bias etc.

RESPONSE: The majority of the analyses presented here were pre-registered to encourage a hypothesis-driven, planned investigation of the data. In particular, we hypothesised that childhood socioeconomic conditions would moderate the IQ-mortality association, in line with previous work, with stronger associations among high SEP individuals.

We believe that the reviewer is referring to the interpretation of the results from the IQ-mortality analysis within each social class. Here, those from higher socioeconomic backgrounds showed no significant IQ-mortality association whereas those from lower socioeconomic backgrounds showed a strong association. This may seem like evidence for a moderator effect, however we do not interpret it as such. Firstly, there was no significant moderator effect in the whole-sample model where the statistical power is much greater. Secondly, we do not explicitly compare the sub-groups in this part of the analyses, so cannot use the results to comment on relative strength of association. Thirdly, as we note in the Results section, the higher social class subgroups are fairly small, particularly the professional subgroup (N = 46). Fourthly, the direction of this moderator effect is contrary to our prediction, with the IQ-mortality association seemingly stronger among those of lower socioeconomic backgrounds. We therefore decide not to over-interpret these findings, and to conclude no evidence for a moderator effect. We have clarified this position in the Discussion. [Page 16, paragraph 2; Page 17, paragraph 1]

In addition, we have amended the Results section to emphasise the effect sizes where appropriate. [Page 12-16]

12. Are the study limitations discussed adequately?

It should be discussed if childhood cognitive function and childhood socioeconomic position could suffer from multicollinearity and how this may affect the interpretation of the results.

RESPONSE: The mutually-adjusted model (without the interaction terms) does not show multicollinearity issues when examining VIFs [see Page 15, paragraph 1]. Furthermore, the correlation between MHT scores and father's occupational social class was small, rather than strong (Spearman's $Rho = 0.19$)[Page 9, paragraph 4]. We therefore do not consider there to be an issue specifically for these two measures. We have clarified this in the discussion. [Page 17, paragraph 3] Paternal occupational class is used as a proxy of childhood SEP. Although this is a well-recognized indicator of childhood SEP, it holds strengths and disadvantages that should be discussed a little further:

- Back in time the breadwinner of the family was most likely to be the father and thus for the majority of the population father's occupational class has been a reasonably good indicator of childhood SEP. But how does it influence the results that it is not possible to categorize for example children of single mothers?
- How do the authors expect the limitations (such as discussed p. 16, l. 26-42) of this measure to influence their findings?

RESPONSE: We agree that father's social class is a historically useful indicator of socioeconomic circumstances and that it may not be as useful in more modern cohorts. We also agree that removing individuals with missing father's social class likely biases the analytic sample, even though deceased/absent/long-term unemployed fathers were still coded according to their most recent

occupation wherever possible. We have acknowledged these points in the paragraph about limitations. [Page 18, paragraph 3]

P. 11, l. 3-19: From the comparison of excluded and included individuals the authors conclude that only a minor difference in cognitive performance exists and that there is no difference in social class. I question this interpretation to be valid for the full sample. As far as I understand from the methods section the exclusions were due to missing cognitive performance score, or social class, or both. Thus, the reported numbers must be based on those with available information on one of the two variables ignoring those with missing information on both. I question that individuals with missing information on both (or all) key variables are comparable to those that are only missing information on one.

RESPONSE: We agree that those removed individuals constitute an interesting sample. Removed individuals had missing childhood cognitive ability, vitality status, father's social class or date of death (in order of size). As noted in the Methods section, removed individuals could have multiple missing variables. As the reviewer points out, the retained vs. removed comparison is based on those who have at least one variable (e.g., social class of retained individuals is compared with removed individuals who have a complete social class). We have clarified this in the Results section [Page 12, paragraph 1] and in the Discussion [Page 18, paragraphs 2 and 3]

P. 15, l. 40-42: Please elaborate on how a dichotomization of social class may overestimate its effect on mortality.

RESPONSE: We have clarified the disadvantages associated with dichotomising social class: "Such a coarse measure of occupational social class may overestimate its contribution to all-cause mortality risk by shifting the estimates for each category towards the mean of included classes – downwards for lower social classes and upwards for higher social classes. Furthermore, it assumes no differences between social classes within each category; here, the mortality risk associated with skilled, intermediate and professional occupational classes was very different." [Page 17, paragraph 2]

P. 16, l. 12: The authors state that the study is based on a nationally representative sample. I question this as I believe a selection into the study exists. The 'complete cases' approach (i.e. including only individuals with full information on all key variables) excludes a number of study participants who did for example not complete the cognitive test or did not have information on paternal occupational class. Moreover, paternal occupational class was coded missing if the father was 'without occupation' or 'unemployed', and finally, it is not clear how children of single mothers were handled. I find it reasonable to argue that children of unemployed fathers and single mothers must represent a socially vulnerable group, especially back in time when this cohort was born. A fairly large number of individuals were excluded due to missing information on childhood cognitive ability. What was the reasons for this? To the reader this is important information in order to evaluate the probability of this being completely random or not. I am not convinced that the excluded individuals are comparable to those included, nor to the population in general. I believe that the paper can be markedly improved by a more thorough analyses of the potential selection bias affecting the results. This could be done by means of inverse probability weighing or at least a more thorough analysis of the potential bias in order to quantify the potential impact on the findings.

RESPONSE: We agree with the reviewer that while the initial sample (The 36-Day Sample) was a nationally-representative cohort the analytic sample is not. We also agree that it is important to highlight to the reader what bias excluded individuals may bring, particularly those excluded due to missing social class. We have made a number of changes to improve how we do this.

As suggested, we have conducted inverse probability weighting to adjust for selection bias (additionally using the information from the 'missing' social class group to inform the weights) and have re-run the survival analyses accordingly. Notably, there has been no substantial change in the reported statistics. We have also added an explicit test of the IQ-mortality association within those with 'missing' social class (Table 3) and have flagged it as an addition to our pre-registered analyses. [Page 15, paragraph 2]

Furthermore, we have also developed the discussion of potential selection bias and its effects in the Discussion. [Page 18, paragraphs 2 and 3]

P. 16, l. 19-42: I agree with the authors that a limitation of life course studies is that the generalizability to younger generations can be compromised by societal changes across time. Also, a specific (categorization of a) measure of occupational class may not suit data from different time periods equally good, as the authors describe. However, I find it unclear whether the authors merely suggest that studies of younger cohorts need to use other measures of occupational class to ensure sensitivity to detect social inequality or if they believe that social class will play no (or only a minor) role in the relationship between cognitive function and mortality in younger cohorts.

RESPONSE: We suggest both the measure – father’s occupational social class – may not be useful in modern cohorts and that occupational social class may be less useful in general. Father’s occupational social class is likely to be less useful a measure in more modern cohorts where both parents are economically active, and occupational social class in general may have a different meaning for modern cohorts where the range of occupations differs. We have discussed this in more detail in the Discussion section. [Page 18, paragraph 3]

Additional comments

To improve readability of the paper, each key concept (i.e. socioeconomic position and childhood cognitive ability) should be given one name based on a conceptually sound reflection and these names should be used consistently throughout the paper.

RESPONSE: We use ‘childhood cognitive ability’ to refer to cognitive ability measured in childhood, and ‘early-life cognitive ability’ to refer to cognitive ability measured in childhood or early-adulthood. ‘IQ scores’ refer to the specific measure used in the analyses (i.e., the indicator of childhood cognitive ability). Similarly, we use ‘socioeconomic circumstances’ to refer to the broader concept of socioeconomic origins and ‘father’s social class’ to refer to the specific indicator used in the analyses. We have amended the text throughout to ensure consistency.

Number of decimals and ways of reporting p-values should be reasonable and consistent throughout the paper.

RESPONSE: We have amended the reporting to be consistently be two decimal places, but retaining the convention of ‘ $p < 0.001$ ’ where appropriate.

VERSION 2 – REVIEW

REVIEWER	Stephen Gilman NIH, USA
REVIEW RETURNED	23-Aug-2020

GENERAL COMMENTS	I appreciate the authors' responses to my earlier comments and think the revised manuscript uniquely contributes to our knowledge about life course pathways which may increase risk for mortality.
---

REVIEWER	Else Foverskov University of Copenhagen, Denmark
REVIEW RETURNED	14-Aug-2020

GENERAL COMMENTS	I think the authors have done a great job revising the manuscript and I do not have further comments.
---

REVIEWER	Gitte Lindved Petersen Translational Type 1 Diabetes Research, Steno Diabetes Center Copenhagen & Section of Epidemiology, Department of Public Health, University of Copenhagen, Denmark
REVIEW RETURNED	25-Aug-2020

GENERAL COMMENTS	Thank you for the opportunity to review the revised manuscript. I believe that the manuscript has been improved after the first round of revisions and the authors have done a good job including informative extra results. I have a few additional remarks that I would like the authors to address. 2. Is the abstract accurate, balanced and complete? The p-values in the abstract are reported with different number of decimals. 3. Is the study design appropriate to answer the research question? Adjustment for sex: The authors justify the adjustment for sex in the main analyses by 1) the well-established association between sex and mortality risk, and 2) by the relevance of being consistent with previous research. I do, however, not find sex to represent a confounder. Unless sex is determining childhood cognitive ability, adjustment for sex is unnecessary as it does not bias the causal relation between exposure and outcome (as for example described by Schisterman and colleagues in 'Overadjustment Bias and Unnecessary Adjustment in Epidemiologic Studies', Epidemiology. 2009 July; 20(4): 488–495). Thus, the unadjusted results should be comparable to previous studies since adjustment merely improves precision of the result estimates. In the present version of the manuscript it is unclear whether the authors believe sex to be a confounder or an effect modifier of the assessed relationship as it is treated as both. The hypotheses behind are quite different and I believe that a clearer argumentation would make the manuscript more informative. Confounder selection and potential unmeasured confounders: As a reader I would prefer the argumentation for confounder selection for the present study to be found within the paper rather than having to investigate previous publications. I will, however, leave it to the editor to decide on this. I agree with the authors that analyses should not be adjusted for covariates such as educational level or adult social class. This would result in overadjustment bias because such factors are on the causal pathway. Consequently, education and socioeconomic circumstances in adulthood do not represent confounders (as stated p. 17, paragraph 2, line 18) but mediators. I suggest the authors to rephrase. My previous comment regarding potential unmeasured confounding was related to potential factors in early life that might affect both cognitive ability and mortality – as the authors note p. 19, paragraph 1. It could benefit the reader if the authors briefly discussed how unmeasured confounding might affect the results (anticipated direction of impact on estimates and whether it is plausible that unmeasured confounding could entirely explain the observed association). Simple quantitative bias analysis (available tools: https://sites.google.com/site/biasanalysis/Home) or calculation of the E-value could be used for such (tool: https://www.evalue-calculator.com/). 10. Are they presented clearly?
---

	I find the presentation of the supplementary results from the additive hazard models a little disturbing (see my previous comment to table 2 for details). I suggest that the authors present the results similarly to those from the Cox models. 11. Are the discussion and conclusions justified by the results The authors state that the lower number of deaths associated with higher IQ and father's social class is 'small' and that the results from the additive hazards approach 'is in keeping with the proportional hazards approach' (in the supplementary section). The focus is still mainly on statistical significance and little on trends and effect sizes (all limitations considered). I miss a discussion of what the results from the supplementary section add to what we see from the main analyses (the absolute versus relative results). In the last paragraph on page 19 the authors discuss the public health implications of their findings. On one side the findings are stated to be important for public health policy and interventions, while on the other the authors point to other potential explanations for why childhood social circumstances do not modify the relationship between cognitive ability and mortality. I suggest omitting the advices for public health policy and interventions if the authors believe that the findings merely can be ascribed to the applied indicators of the construct of interest. Instead, a balanced concluding remark could be added on how much we should believe in a causal relationship and how strong the effect appears to be in a public health perspective.
--	--

VERSION 2 – AUTHOR RESPONSE

Reviewer 3

Thank you for the opportunity to review the revised manuscript. I believe that the manuscript has been improved after the first round of revisions and the authors have done a good job including informative extra results. I have a few additional remarks that I would like the authors to address.

2. Is the abstract accurate, balanced and complete?

The p-values in the abstract are reported with different number of decimals.

RESPONSE: We would like to thank the reviewer for pointing this out. The Abstract has now been updated to match the reporting style of the main text.

3. Is the study design appropriate to answer the research question?

Adjustment for sex:

The authors justify the adjustment for sex in the main analyses by 1) the well-established association between sex and mortality risk, and 2) by the relevance of being consistent with previous research. I do, however, not find sex to represent a confounder. Unless sex is determining childhood cognitive ability, adjustment for sex is unnecessary as it does not bias the causal relation between exposure

and outcome (as for example described by Schisterman and colleagues in ‘Overadjustment Bias and Unnecessary Adjustment in Epidemiologic Studies’, *Epidemiology*. 2009 July; 20(4): 488–495). Thus, the unadjusted results should be comparable to previous studies since adjustment merely improves precision of the result estimates. In the present version of the manuscript it is unclear whether the authors believe sex to be a confounder or an effect modifier of the assessed relationship as it is treated as both. The hypotheses behind are quite different and I believe that a clearer argumentation would make the manuscript more informative.

RESPONSE: We appreciate the reviewer’s feedback on the role of sex in the analyses. As the reviewer mentions, we include sex as a covariate to address the sex-mortality association and to be consistent with previous work. While previous work has shown no significant mean differences in childhood IQ between boys and girls, boys are over-represented at the higher and lower extremes of the IQ distribution (Deary, Thorpe, Wilson, Starr & Whalley, 2003). We intend to treat sex as a confounding factor, as in previous studies investigating the IQxSES interaction. We note that adjusting for sex as a confounder does not impact on the IQ-mortality or SES-mortality associations.

We have noted that treatment of sex as a confounder in the Analysis section of the Methods (Page 8, line 15). We have also added unadjusted, sex-adjusted and age-adjusted analyses of the interaction effect in the Supplementary Material.

Confounder selection and potential unmeasured confounders:

As a reader I would prefer the argumentation for confounder selection for the present study to be found within the paper rather than having to investigate previous publications. I will, however, leave it to the editor to decide on this.

RESPONSE: We believe that the selection of these particular confounders is relatively common, and we do briefly touch upon – for example – sex differences in IQ, social class and mortality rate. We have also signposted readers to relevant studies when discussing the confounders and the potential for unmeasured confounding.

I agree with the authors that analyses should not be adjusted for covariates such as educational level or adult social class. This would result in overadjustment bias because such factors are on the causal pathway. Consequently, education and socioeconomic circumstances in adulthood do not represent confounders (as stated p. 17, paragraph 2, line 18) but mediators. I suggest the authors to rephrase.

RESPONSE: We agree that as factors down-stream from childhood cognitive ability and social class, education and adult socioeconomic circumstances are better described as mediators. We have amended the relevant sentences. (Page 17, paragraph 2, line 16-20):

“Although we only included sex as a potential confounder, a more thorough investigation of bias using other potential confounders and mediators is available elsewhere (9–11), including in the same Scottish Mental Survey 1947 subsamples (6,7,14,15). Notably, childhood cognitive ability and childhood socioeconomic circumstances have been shown to predict later-life health, even after accounting for mediators such as education and socioeconomic circumstances in adulthood (5,6,25,27).”

My previous comment regarding potential unmeasured confounding was related to potential factors in early life that might affect both cognitive ability and mortality – as the authors note p. 19, paragraph 1.

It could benefit the reader if the authors briefly discussed how unmeasured confounding might affect the results (anticipated direction of impact on estimates and whether it is plausible that unmeasured confounding could entirely explain the observed association). Simple quantitative bias analysis (available tools: <https://sites.google.com/site/biasanalysis/Home>) or calculation of the E-value could be used for such (tool: <https://www.evalue-calculator.com/>).

RESPONSE: As we discuss elsewhere in the Discussion (Page 17, lines 16-19), previous work has shown that the association between childhood cognitive ability and mortality risk remains once a variety of potential confounders are accounted for. We have added more specific discussion of the impact of unmeasured confounders to the relevant section of the Discussion (Page 19, lines 7-9), including how socioeconomic conditions at birth may impact both childhood cognitive ability and mortality risk and further previous work that has accounted for potential confounders.

We thank the reviewer for suggesting the sensitivity analysis tools for examining confounding. As suggested, we have included E-value analysis in the Supplementary Material as suggested. To summarise here, based on HRs and CIs from Table 2, any potential confounder would have to have a relatively strong association (in the context of the associations reported in this study) with both IQ and mortality risk to fully explain their observed association. We have added reference to the E-value analysis and noted that a relatively strong unmeasured confounder would be required, in both the Results (Page 13, paragraph 1, lines 11-13) and the Discussion (Page 19, paragraph 1, lines 20-23).

10. Are they presented clearly?

I find the presentation of the supplementary results from the additive hazard models a little disturbing (see my previous comment to table 2 for details). I suggest that the authors present the results similarly to those from the Cox models.

RESPONSE: We have amended the Table A3 (previously Table A2) in the Supplementary Material to remove the Interaction column and have instead described the interaction effect in-text (Page 4, paragraph 3). We have also ensured that the new Table A1 does not describe both the main and interaction effects for the final model.

11. Are the discussion and conclusions justified by the results The authors state that the lower number of deaths associated with higher IQ and father's social class is 'small' and that the results from the additive hazards approach 'is in keeping with the proportional hazards approach' (in the supplementary section). The focus is still mainly on statistical significance and little on trends and effect sizes (all limitations considered). I miss a discussion of what the results from the supplementary section add to what we see from the main analyses (the absolute versus relative results).

RESPONSE: We have included reference to the relatively small magnitude of the association and to the 95% CIs including a zero value in the Additive Hazards analyses (Supplementary Material, Page 4, paragraph 3).

"We then tested the moderating effect of father's social class. The IQ-social class interaction predicted a small increase of 0.13 deaths per 10,000 person years, though this was only marginally-significant and 95% CIs included 0 (95% CI [0.00-0.25], $p = 0.05$). That is, a 1SD advantage in childhood cognitive ability predicted 0.13 more deaths per increase in father's social class. Note that this association was much smaller in magnitude than the decreases in additional deaths associated with the main effects."

We further include discussion of the by-class trend in the number of additional deaths predicted by a 1SD advantage in IQ, namely that the benefit appears to diminish as social class increases. We note, however, that the interaction effect was not significant and that the confidence intervals for some social class groups are very wide (Supplementary Material, Page 4, paragraph 4). We have also included more reflection on the relative results from the Additive Hazards analyses at the end of the section (Supplementary Material, Page 4, paragraph 5).

“We conclude that the additive hazards approach presented here is in keeping with the proportional hazards approach used in the main text. In the context of the underlying mortality hazard for this sample, advantage in IQ and childhood social class predict modest reductions in the number of deaths per 10,000 person years. In contrast, the interaction effect was relatively weak, with no consistent trend across classes.”

We have also added reference to the additive regression analyses in the main text, noting that they complement the Cox regression results. (Page 16, paragraph 2, lines 18-22)

“Examination of the absolute difference in the risk of mortality likewise demonstrated that higher childhood cognitive ability and higher childhood socioeconomic circumstances significantly predicted fewer deaths per 10,000 person years, and that the interaction did not significantly predict additional deaths (Supplementary Material).”

In the last paragraph on page 19 the authors discuss the public health implications of their findings. On one side the findings are stated to be important for public health policy and interventions, while on the other the authors point to other potential explanations for why childhood social circumstances do not modify the relationship between cognitive ability and mortality. I suggest omitting the advices for public health policy and interventions if the authors believe that the findings merely can be ascribed to the applied indicators of the construct of interest. Instead, a balanced concluding remark could be added on how much we should believe in a causal relationship and how strong the effect appears to be in a public health perspective.

RESPONSE: In the concluding statement we summarise that we found no evidence that this particular measure of material socioeconomic circumstances in childhood (father’s social class) moderates the IQ-mortality association, and that it is important to address childhood cognitive ability as a separate source of later health inequality. However, we also acknowledge that this may not apply to other measures of socioeconomic circumstances (e.g., parental education) which are not available here. The intention was to provide a recommendation based on our (and previous) results while making the reader aware of an important caveat. However, we appreciate that this provides a conflicting message. We have moved mention of other measures of socioeconomic circumstances as moderators to earlier in the Discussion, where we discuss the limitations of the ‘father’s social class’ proxy measure. We have also clarified that future studies should focus on identifying causal mechanisms linking childhood IQ/SES to mortality risk (using regression discontinuity, etc.), as the present study supplies evidence only of temporal causality. The final sentences (Page 20, paragraph 1, lines 1-8) now read:

“This is an important consideration for public health policy and interventions: childhood cognitive ability is a source of health inequality independent from material socioeconomic resources in childhood, and needs to be addressed separately. Future work should focus on building causal evidence and clarifying the mechanisms by which childhood cognitive ability and childhood socioeconomic conditions affect mortality risk.”

VERSION 3 – REVIEW

REVIEWER	Gitte Lindved Petersen Department of Translational Type 1 Diabetes Research, Steno Diabetes Center Copenhagen, Denmark & Section of Epidemiology, Department of Public Health, University of Copenhagen, Denmark
REVIEW RETURNED	10-Nov-2020
GENERAL COMMENTS	The authors have added valuable details to the manuscript. I have no further comments.